# Gut Microbiota Changes during Dimethyl Fumarate Treatment in Patients with Multiple Sclerosis

**DOI:** 10.3390/ijms24032720

**Published:** 2023-02-01

**Authors:** Caterina Ferri, Massimiliano Castellazzi, Nicola Merli, Michele Laudisi, Elisa Baldin, Eleonora Baldi, Leonardo Mancabelli, Marco Ventura, Maura Pugliatti

**Affiliations:** 1Department of Neuroscience and Rehabilitation, University of Ferrara, 44121 Ferrara, Italy; 2Department of Neuroscience and Rehabilitation, St. Anna University Hospital, 44124 Ferrara, Italy; 3Interdepartmental Research Center for the Study of Multiple Sclerosis and Inflammatory and Degenerative Diseases of the Nervous System, University of Ferrara, 44121 Ferrara, Italy; 4IRCCS Istituto delle Scienze Neurologiche di Bologna, 40139 Bologna, Italy; 5Department of Medicine and Surgery, University of Parma, 43124 Parma, Italy; 6Microbiome Research Hub, University of Parma, 43124 Parma, Italy; 7Laboratory of Probiogenomics, Department Chemistry, Life Sciences and Environmental Sustainability, University of Parma, 43124 Parma, Italy

**Keywords:** gut microbiota, multiple sclerosis, dimethyl fumarate, gastrointestinal side effects, flushing, *Clostridium*

## Abstract

The gut microbiota is involved in the development of the immune system and can modulate the risk for immune-mediated disorders such as multiple sclerosis (MS). Dysbiosis has been demonstrated in MS patients and its restoration by disease-modifying treatments (DMTs) is hypothesized. We aimed to study the changes in gut microbiota composition during the first 6 months of treatment with dimethyl fumarate (DMF), an oral DMT, and to identify the microorganisms associated with DMF side effects. We collected and analyzed the gut microbiota of 19 MS patients at baseline and after 1, 3, and 6 months of DMF treatment. We then cross-sectionally compared gut microbiota composition according to the presence of gastrointestinal (GI) symptoms and flushing. Overall, the gut microbiota biodiversity showed no changes over the 6-month follow-up. At the genus level, DMF was associated with decreased *Clostridium* abundance after 6 months. In subjects reporting side effects, a higher abundance of *Streptococcus*, *Haemophilus*, *Clostridium*, *Lachnospira*, *Blautia*, *Subdoligranulum*, and Tenericutes and lower of *Bacteroidetes*, *Barnesiella*, *Odoribacter*, *Akkermansia*, and some Proteobacteria families were detected. Our results suggest that gut microbiota may be involved in therapeutic action and side effects of DMF, representing a potential target for improving disease course and DMT tolerability.

## 1. Introduction 

The “gut-brain axis” is a bidirectional communication network linking the central nervous system (CNS) and the gastro-enteric tract. It is mediated by nutrients and neuroendocrine, metabolic, and immunological signals which modulate the inflammatory response and regulate immune homeostasis [1,2]. The gut microbiota is involved in the maintenance of health homeostasis through a number of pivotal structural and metabolic functions, including those on the CNS and immune response development and maturation [1], and an alteration of the gut colonization can increase the risk for immune-mediated disorders [2,3]. Despite an increasing interest in the gut–brain axis for therapeutic implications, the study of the gut microbiota is limited due to a high interindividual variability [4] and its interaction with several environmental and/or lifestyle exposures [5,6].

Multiple sclerosis (MS) is an inflammatory and degenerative disease of the CNS, potentially highly disabling and most frequently affecting young adults. It is a multifactorial condition with environmental and genetic factors interacting with one another at a susceptible age to generate inappropriate immune-mediated or autoimmune responses against CNS myelin [7]. Evidence from MS animal models suggests a pivotal role of the gut microbiota in developing the disease [8]. MS patients have been shown to feature a different composition of gut microbiota compared to healthy subjects [8,9,10,11,12,13]. However, to date, no specific gut microbiota composition has been reported in association with MS nor the predisposing role of specific gut bacteria in modulating the risk for the disease. Furthermore, MS disease-modifying treatments (DMTs) can affect the abundance of gut commensal bacteria [9,10,11,14,15,16] but the effect of DMT-induced changes on the gut microbiota remains to be elucidated as the evidence is poor and mainly based on cross-sectional comparisons studies, with the exception of only three prospective studies [10,15,16].

Dimethyl fumarate (DMF), the methyl ester of fumaric acid, is an oral drug for the treatment of relapsing-remitting MS (RRMS) ultimately promoting anti-inflammatory and neuroprotective antioxidant effects by targeting and activating the Nrf2 transcription pathway [17]. A complementary action of DMF is the inhibition of the transcription factor NFkB [18,19]. DMF has been hypothesized to affect the gut–brain axis, increasing the abundance of bacteria producing short-chain fatty acids (SCFAs), reducing intestinal barrier permeability, and exerting an antimicrobial effect [20,21,22]. Furthermore, DMF-induced lymphopenia has been associated with a distinct microbial profile at baseline [16] and a link between DMF gastrointestinal (GI) side effects and changes in the gut microbiota has been hypothesized [15], although no evidence has ever been reported.

As an attempt to provide a contribution to unraveling the relationship between MS, gut microbiota composition, and its interaction with DMF, we aimed to detect and characterize changes in the gut microbiota in RRMS patients during DMF treatment.

## 2. Results

### 2.1. Study Population Characteristics

We included 19 RRMS patients screened to start treatment with DMF. The main demographic, clinical, and lifestyle characteristics of the study population are shown in Table 1.

Seven patients were previously treated with interferon β1a (*n* = 5), interferon β1b (1), or Glatiramer acetate (1). The reasons for switching to DMF were intolerance (2), the ineffectiveness of previous DMTs (2), or patients’ requests for an oral drug (3).

Immune-mediated comorbidities in the study population included thyroiditis (4), psoriasis (2), undifferentiated connective tissue disease (1), lichen planus (1), and vitiligo (1). One patient referred to a history of Henoch Schonlein purpura, one had hyperinsulinism, and two patients suffered from psychiatric disorders (depression, bipolar disorder). Only one patient was taking proton pump inhibitors during the study start and five women were under oral contraceptives. Four subjects were taking vitamin D.

The overall diet composition (Appendix A) and other lifestyle factors (physical activity, sun exposure, smoking habit) did not change significantly between the baseline and over the 6-month study. Over the observation period, no clinical relapses or MRI activity were detected.

### 2.2. Gut Microbiota Analysis

Available stool samples were 19 at baseline, 18 at month 1, 19 at month 3, and 17 at month 6. The analysis was therefore performed on a total of 73 samples.

A high interindividual variability in the gut microbiota composition was detected at baseline, with Bacteroidetes (49%) and Firmicutes (43%) being the most represented phyla (Figure 1).

The gut microbiota biodiversity expressed with the Chao 1 index did not significantly change from the baseline and during DMF treatment (Table 2). Overall, the taxonomic analysis did not show relevant changes in the Bacteroidetes/Firmicutes (B/F) ratio. However, in silico analyses of the data showed a transient reduction of the phylum Proteobacteria (*p* = 0.014) at month 1. Among families, only Clostridiaceae showed a significant modification of the relative abundance, which was reduced at month 6 of follow-up (*p* = 0.006). At the genus level, we found a transient increase of *Anaerostipes* (*p* = 0.021) at month 1 and of Ruminococcaceae UCG002 (*p* = 0.027) at month 3, while a reduction of Ruminococcaceae NK4A214 group was observed at month 3 (*p* = 0.047). The only change at month 6 from the baseline was a significant decrease in the genus *Clostridium* sensu strictu 1 (*p* = 0.006), reflecting the decrease of the family Clostridiaceae. The species and subspecies of the genus *Bifidobacterium* did not significantly change over the follow-up period compared to the baseline (Appendix A).

### 2.3. Gut Microbiota Composition and DMF Side Effects

GI side effects tended to decrease over the follow-up period from treatment start with DMF, whereas the frequency of flushing remained stable and the prevalence of lymphopenia, as expected, increased with treatment duration (Table 3). Significant differences in the microbiota composition by DMF-related side effects are reported at months 1, 3, and 6 after treatment initiation.

Among patients with GI side effects, a lower abundance of the family Bacteroidaceae and genus *Bacteroides* (*p* = 0.033) and a higher abundance of Tenericutes (*p* = 0.037), Streptococcaceae (*p* = 0.021), *Streptococcus* (*p* = 0.013), and *Subdoligranulum* (*p* = 0.026) were observed at month 1. At month 3, lower abundances of Barnesiellaceae (*p* = 0.022), *Barnesiella* (*p* = 0.013), and *Odoribacter* (*p* = 0.033), and higher abundances of Clostridiaceae (*p* = 0.031), *Clostridium* (*p* = 0.039), and *Blautia* (*p* = 0.028) were associated with GI symptoms. At month 6, while GI side effects almost completely disappeared, they were still associated with lower levels of Proteobacteria (*p* = 0.023), Acidaminococcaceae (*p* = 0.017), and Burkholderiaceae (*p* = 0.032), and a non-significant tendency to higher Clostridiaceae and *Clostridium* abundances (*p* = 0.051) (Table 4).

Flushing was only associated with a higher abundance of *Paraprevotella* at month 1 (*p* = 0.025). At month 3, the same *Paraprevotella* appeared to be lower in association with flushing (*p* = 0.047), a similar but nonsignificant tendency was found for Succinivibrionaceae. On the other side, Pasteurellaceae (*p* = 0.029), *Haemophilus* (*p* = 0.024), and *Streptococcus* (*p* = 0.011) were more represented in subjects with flushing at month 3; the Lachnospiraceae NK4A136 group and *Faecalibacterium* showed a similar tendency in association with flushing but without reaching statistical significance (*p* = 0.050). Finally, at month 6 Enterobacteriaceae (*p* = 0.021) and *Akkermansia* (*p* = 0.036) were higher in subjects without flushing. Similarly to the results at month 3, Pasteurellaceae and *Haemophilus* (*p* = 0.025) appeared higher in association with flushing, and so did *Lachnospira* (*p* = 0.011). Clostridiaceae and *Clostridium* were also more abundant in subjects with flushing even if without significance (*p* = 0.056) (Table 5).

Considering the low prevalence of lymphopenia up to month 6, we were not able to assess microbiota differences according to it.

## 3. Discussion

Through the collection of four fecal samples per patient over a 6-month observation period, we found that the gut microbiota α biodiversity was not significantly affected by treatment with DMF, whereas some changes were observed after taxonomic analysis. At the phylum level, a transient reduction of Proteobacteria was detected at month 1. At the family and genus levels, a strong reduction of the Clostridiaceae 1 family members, specifically of the genus *Clostridium*, was observed at month 6 after treatment initiation. In addition, an increase of *Anaerostipes* at month 1 and changes in some genera belonging to the Ruminococcaceae family at month 3 were detected. The relative abundance of the *Bifidobacterium* species and subspecies was instead not affected by DMF assumption.

The existing literature on the effect of DMTs on gut microbiota composition is scarce and mostly based on cross-sectional study designs (Table 6), potentially biased by high interindividual variability and environmental confounders. Even considering the different study designs, our results are in line with previous evidence highlighting the absence of impact on biodiversity by DMTs [9,10,11,14,15,16], looking instead at the effect of DMF on taxonomic analysis, our results are quite conflicting with previous studies. A cross-sectional study conducted in the Northern American population suggested a lower abundance of the phyla Firmicutes and Fusobacteria and of Clostridiales order (families Lachnospiraceae and Veillonellaceae) in patients taking DMF compared to treatment-naïve subjects [14]. The longitudinal effect of DMF on gut microbiota composition has only been investigated by two studies with 3-month follow-ups, one with three samplings per patient and the other with only two [15,16].

Storm-Larsen and colleagues found a reduction of the phylum Actinobacteria after 2 weeks from DMF start (mainly driven by *Bifidobacterium*) and an increase of Firmicutes after 3 months (mostly driven by *Faecalibacterium*) [15]. Diebold showed instead a decrease in *Coprococcus eutactus* and *Enterococcus gilvus* and an increase in *Lactobacillus pentosus* at the third month of treatment, whereas *Akkermansia muciniphila* tended to decrease non-significantly [16]. Due to the different time points used, our findings can only be compared with those of Storm-Larsen and Diebold for the analysis at month 3 from DMF start.

Our finding on *Clostridium* reduction could be explained by the property of DMF to inhibit the in vitro growth of the epsilon toxin (ETX) producer *Clostridium perfringens* [22]. The Clostridiales order has received particular attention in the field of demyelinating diseases. *Clostridium perfringens* is an anaerobic bacillus that can be classified into five subclasses based on the type of exotoxin produced. Type A usually colonizes the human intestine while types B and D, which produce ETX, colonize mostly the intestines of ruminant animals. ETX is a strong neurotoxin secreted as an inactive precursor; after the cleavage, it crosses the intestinal barrier, enters the bloodstream, and binds to receptors on the surface of brain endothelial cells, where oligomerizes to form a heptameric pore, responsible for increased permeability of the blood–brain barrier [23]. ETX is also cytotoxic in vitro for a subset of astrocytes and microglia cells and is capable of acting directly on oligodendrocytes, without forming pores, potentially triggering demyelination [23,24]. In humans, increased immunoreactivity against ETX has been described in serum and cerebrospinal fluid of the MS population [25,26], suggesting a role of *Clostridium perfringens* in MS pathogenesis. Furthermore, this microorganism has been found to be overabundant in Neuromyelitis Optica, another inflammatory disease of the CNS [27]. Of note, the ETX model has been proposed among complementary pre-clinical “Inside-out” models of MS in order to develop new therapeutic strategies [28]. Even if no specific species of *Clostridium* could be identified in our study, the observed overall reduction of genus *Clostridium* in MS patients treated with DMF may support the role of this microorganism as a therapeutic target for MS.

It has been shown that other members of the order Clostridiales, belonging to the families Lachnospiraceae and Ruminococcaceae, can inhibit oligodendrocyte differentiation in mice prefrontal cortex through the action of their metabolites on gene expression and result in mice social behavior modulation [29]. No direct association has been proposed between these families and MS pathogenesis; evidence suggests they could be reduced in subjects taking DMF [14] but they also appeared less abundant in MS pediatric patients compared to healthy controls [11]. It is, therefore, hard to draw conclusions on our findings concerning the Lachnospiraceae and Ruminococcaceae genera modifications at months 1 and 3.

The phylum Proteobacteria, which includes various pathogenetic microorganisms such as *Escherichia*, *Shigella*, *Klebsiella*, and *Haemophilus*, has been associated with dysbiosis, epithelial dysfunction, and pathological conditions, although not with MS [30]. In our study population, the number of Proteobacteria appeared to decrease at month 1 after DMF start, which could be intended as a beneficial effect; however, it was not confirmed at months 3 and 6. 

To the best of our knowledge, no analysis on *Bifidobacterium* species and subspecies has ever been performed in relation to the effect of DMTs, and ours is novel evidence. These microorganisms are crucially relevant during the early stages of life, highly abundant in childhood and declining in adulthood [31]. Wagenfeld and colleagues investigated the bifidobacterial composition in MS patients through culture analysis [32]. They suggested a lower abundance of *Bifidobacterium adolescentis* in 17 MS patients compared to controls. Whereas, *Bifidobacterium longum* and *Bifidobacterium bifidum* did not differ in the two groups. Remarkably, *Bifidobacterium adolescentis* has been hypothesized to exert a beneficial effect on the immune response in MS [32]. Although the role of *Bifidobacterium* species and subspecies in MS remains to be elucidated, our findings support their stability during DMF treatment.

The comparison of the gut microbiota composition of subjects with and without DMF side effects yielded some differences according to the occurrence of GI symptoms and flushing. At month 1, GI symptoms were associated with higher levels of *Streptococcus, Subdoligranulum*, and Tenericutes phylum, while *Bacteroides* were lower compared to asymptomatic patients. At month 3, higher levels of *Clostridium* and *Blautia* and lower levels of *Barnesiella* and *Odoribacter* were instead observed in patients with GI symptoms. At month 6, when the prevalence of GI symptoms was—as expected—reduced, lower abundances of Proteobacteria, Burkholderiaceae, and Acidaminococcaceae and a trend towards higher levels of *Clostridium* were detectable among symptomatic patients. The most consistent and persistent change we observed in subjects with DMF-related flushing was the higher abundance of Pasteurellaceae and *Haemophilus* over the whole follow-up period, statistically significant at months 3 and 6. In addition, higher abundances of *Streptococcus* and *Lachnospira* were detected in association with flushing at months 3 and 6, respectively, while *Akkermansia* and Enterobacteriaceae appeared more represented in the group without flushing. Storm-Larsen and colleagues reported an association between DMF-related GI effects and a lower level of *Bacteroides* at baseline and a higher abundance of *Dialister* after 2 weeks from treatment initiation [15]. No data on flushing by DMF and microbiota have ever been published. Diebold and colleagues described instead a baseline microbiome signature consisting of the presence of *Akkermansia muciniphila* and contextual absence of *Prevotella copri*, which was predictive of lymphopenia during the treatment [16]. In our population, considering the short-lasting follow-up and the multiple sampling per patient, we decided to carry on a cross-sectional analysis to study the microbiota profile concomitantly with DMF side effects without searching for a baseline microbial profile predictive of side effects. As our findings are, to the best of our knowledge, novel, we have tried to interpret them in the light of current knowledge. Considering the known pathogenetic role of *Clostridium*, *Streptococcus*, and some genera of Proteobacteria, an association between higher abundances of *Haemophilus*, *Clostridium*, and *Streptococcus* and GI symptoms and flushing by DMF should not be ruled out. At the same time, it is not surprising that the genera *Bacteroides*, *Barnesiella*, and *Odoribacter*, belonging to the phylum Bacteroidetes and generally advocating eubiosis, were reduced in association with side effects. Interestingly, inflammatory bowel disease has been associated with a reduction of butyrate-producing bacteria [33], this evidence could further justify the lower levels of the butyrate-producer *Odoribacter* in subjects with GI symptoms during DMF assumption. Conversely, the variable abundance of the genus *Paraprevotella* in subjects with and without flushing according to the time point suggests that it is not directly associated with the side effect. The reduced abundance of *Akkermansia* in subjects with flushing could instead subtend a pathogenetic meaning as this genus comprises species such as *A. muciniphila*, which contributes to the maintenance of a healthy gut barrier [34]. Conversely, the role of the genera *Blautia* and *Subdolingranulum* (belonging to Lachnospiraceae and Ruminococcaceae respectively) which were more abundant in patients with GI symptoms, and those of the Proteobacteria families Enterobacteriaceae, Acidaminococcaceae, and Burkholderiaceae, less abundant in association with flushing or GI symptoms, remains uncertain. Notably, the overabundant or depleted microorganisms were quite specific according to the side effect. Overall, our findings suggest a potential mediating role of some microorganisms on GI symptoms and flushing which could be triggered by dysbiosis, although this relationship deserves further confirmation.

The main limitation of our study is the small sample size. Due to the small number of participants, a multivariate analysis could not be performed to better detect potential confounders. Although we cannot exclude the influence of comorbidities such as autoimmune disorders, concomitant medications, or previous DMTs on the baseline gut microbiota composition, their influence has little relevance in the context of a longitudinal study. Another limitation is that the 6-month study period did not allow us to make any inference on DMF-related long-term changes in gut microbiota composition as well as their role in MS prognosis, clinical relapses, and MRI activity. Finally, the lack of analysis by species may have not allowed us to identify further changes in the microbiota.

Considering that it could be affected by multiple external and internal stimuli, the study of gut microbiota composition is far from simple. A strength of our study is the exclusion of the interference of lifestyle habit modification during the observation period which might otherwise have influenced our results. Potentially, the underestimation of the influence of environmental conditions, mainly diet, on the gut microbiota by previous research could explain conflicting results. Further investigations are needed to improve our understanding of the effect of DMTs, such as DMF, on the gut microbiota and if it could subtend therapeutic mechanisms or side effects presentation. To obtain valid results, environmental factors known to have an influence on the gut microbiota should be considered as potential confounders or effect modifiers.

## 4. Materials and Methods

### 4.1. Study Design and Study Population

We performed a longitudinal prospective study, patients were consecutively recruited at the MS Center, University Hospital of Ferrara, northern Italy, and sampled in the pre-pandemic era. Inclusion criteria were a diagnosis of RRMS according to the 2010 McDonald criteria or the 2017 revised McDonald criteria [35,36], age between 18 and 65 years, and being a candidate for treatment initiation with DMF according to the principle of good clinical practice. Exclusion criteria were current DMF treatment, antibiotic treatment or high-dose corticosteroids within 30 days before study enrollment, having undergone previous GI surgery, current pregnancy, and current or recent (within 12 months before enrollment) treatment with immunosuppressants.

A starting DMF dose of 120 mg orally twice a day for 7 days followed by a maintenance dose of 240 mg orally twice a day was prescribed for each patient.

Information about MS history (year of onset, symptoms and signs, previous treatments if any, other concomitant medications) and comorbidities were recorded for descriptive purposes. A clinical assessment including neurological examination with Expanded Disability Status Score (EDSS) calculation was performed at baseline (i.e., before starting on DMF) and at months 1, 3, and 6 after treatment initiation. Lifestyle exposures (cigarette smoking habit, level of physical activity, sun exposure, and diet) have been collected at baseline and during the follow-up as potential effect confounders. Eating habits were collected using a self-administered questionnaire and entered into an “open access” online platform (*Grana Padano Nutritional Education*, https://www.educazionenutrizionale.granapadano.it/it/ accessed on 23 March 2022) whose outputs allowed us to compute the estimated amount (absolute and percentage values) of diet macro- and micro-nutrients. None of the patients received specific dietary advice during the observational period, we just recommended drug administration after meals. DMF side effects were also recorded at each time point. All the mentioned data were collected by an MS neurologist.

### 4.2. Analysis of the Gut Microbiota

Stool specimens were collected before DMT start (baseline) and at months 1, 3, and 6, respectively. Each patient was provided with Norgen© tubes for the collection and storage of fecal nucleic acids for up to 7 days at room temperature (Norgen Biotek Corp., Thorold, ON, Canada; https://norgenbiotek.com/ accessed on 1 September 2022). The samples were then stored at −20°C at the Laboratory of Neurochemistry, Ferrara University Hospital until processing. The gut microbiota analysis was performed by GenProbio s.r.l. (Probiogenomics Lab, University of Parma, Italy). After extracting bacterial DNA from stool samples, the region V4 of 16S rRNA was amplified through PCR and sequenced using the Illumina MiSeq platform with the 2 × 150 bp paired-end protocol [37]. The generated “reads” were filtered and then grouped into operational taxonomic units (OTUs) through the Quantitative Insight Into Microbial Ecology (QUIIME) 2 software. QUIIME is an open-source software performing microbiome analysis from raw DNA sequencing data, allowing for quality filtering, OTU picking, taxonomic assignment, phylogenetic tree reconstruction, and diversity analyses and visualizations (www.quiime.org accessed on 5 June 2022). *Bifidobacterium* species and subspecies were analyzed using the internal transcribed spacer bifidobacterial profiling (ITSbp) [38]. The database Silva v. 132 and a customized database containing currently known Bifidobacteria sequences were used to obtain the 16S rRNA analysis outputs and bifidobacterial ITS analysis, respectively. The analysis of α biodiversity was performed by computing the Chao 1 index. The taxonomic analysis was conducted at phylum, family, and genus levels. Among all the measured microbes, we considered the most abundant or relevant ones and included 7 phyla, 21 families, 33 genera, and 9 bifidobacterial species and subspecies (Appendix A). The phylum Fusobacteria and some genera were excluded from the statistical analysis as they were missing in almost all of the study population. The two main phyla were also measured as the Bacteroidetes/Firmicutes (B/F) ratio.

### 4.3. Statistical Analysis

Descriptive statistics are presented for continuous (means and 95% confidence interval or standard deviations) and categorical or binomial (frequencies) variables.

The nonparametric Wilcoxon signed-rank test was used to characterize gut microbiota composition longitudinally, at phylum, family, and genus levels, and for the species and subspecies of *Bifidobacterium*. Gut microbiota composition at months 1, 3, and 6 after treatment initiation was compared with baseline composition. We set statistical significance as *p* < 0.05. As for the Wilcoxon signed-rank test, the unadjusted *p*-values were computed in line with previous work [15].

We considered smoking habit, sun exposure, and physical activity as confounding factors; therefore, we compared these lifestyle characteristics at baseline with that during the follow-up to rule out their potential confounding effect on microbiota composition modifications during the observation period.

We also performed a cross-sectional analysis using the Kruskal–Wallis test to compare the microbial composition of patients with and without GI side effects and flushing by DMF.

Statistical analysis was performed with SPSS Statistics for Windows, version 27 (IBM Corporation, Armonk, NY, USA).

## 5. Conclusions

DMF treatment did not significantly modify the gut microbiota biodiversity during the first 6 months of treatment. At the genus level, *Clostridium* abundance decreased after 6 months from treatment start. Considering the potential neurotoxic effect of some *Clostridium* species, such as the ETX producer *Clostridium perfringens*, *Clostridium* suppression could represent a therapeutical mechanism of DMF. Furthermore, a higher or lower abundance of some microorganisms has been associated with specific DMF side effects. The study’s novelty lies in the evidence that the gut microbiota could mediate DMF therapeutic and side effects in MS patients.

These issues deserve further research. A better understanding of the interaction between the gut microbiota and DMTs could have implications for therapeutic efficacy and treatment tolerability.

## Figures and Tables

**Figure 1 ijms-24-02720-f001:**
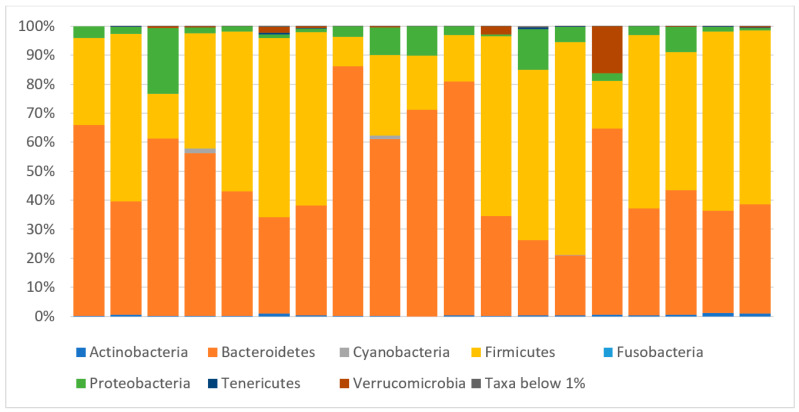
Relative abundance at phyla level for each patient (column) at baseline.

**Table 1 ijms-24-02720-t001:** Main demographic, clinical, and lifestyle characteristics of the study population at baseline.

Demographic Characteristics	*n* = 19
Age, mean (95%CI), range (years)	38.5 (34.6–42.9) 27–55
Women, *n* (%)	13 (68.4)
Clinical Characteristics	
Age at MS onset, mean (95%CI), range (years)	34.1 (29.8–38.2) 16-53
MS duration (years), mean (95%CI), range (years)	6.79 (4.4–9.7) 3–25
Previous treatment with DMT, *n* (%)	7 (36.8)
Number of relapses in the previous 2 years, mean (95%CI), range	1.2 (0.79–1.63) 0–3
New lesions on the baseline MRI, *n* (%)	12 (63.2)
Contrast-enhancing lesions on the baseline MRI, *n* (%)	3 (15.8)
EDSS score, median (range)	1.5 (1–4)
BMI, mean (95%CI)	24.5 (22.6–26.8)
Lifestyle Characteristics	
Smoking	
Ever smokers, *n* (%)	11 (57.9)
Sun exposure	
Sun exposure during the weekend over 2 h, *n* (%)	14 (73.7)
Physical activity (≥once per week)	
Intense physical activity, *n* (%)	8 (42.1)

95%CI: 95% confidence interval; MS: multiple sclerosis; DMT: disease-modifying treatment; EDSS: Expanded Disability Status Scale; MRI: magnetic resonance imaging; BMI: body mass index.

**Table 2 ijms-24-02720-t002:** Biodiversity and relative abundance of the gut microbiota bacteria by phylum, family, and genus during the treatment with DMF. The α biodiversity is expressed as the Chao 1 index. We reported all the analyzed phyla and the families and genera with significant modification in relative abundance during DMF treatment compared to baseline values. The taxa that increased at any time point during DMF treatment are shown in green and those that reduced in orange.

	Baseline *n* = 19	Month 1 *n* = 18	*p* ^a^	Month 3 *n* = 19	*p* ^a^	Month 6 *n* = 17	*p* ^a^
Biodiversity Chao 1, mean (SD)	154.335 (55.8504)	154.0416 (56.02970)	0.420	157.33958 (61.726292)	0.573	157.66045 (58.866644)	0.246
Relative abundance Phylum, mean (SD)							
Actinobacteria	0.00373563 (0.003461430)	0.00257725 (0.002086759)	0.170	0.00297034 (0.003069419)	0.355	0.00327850 (0.004308527)	0.687
Bacteroidetes	0.49090616 (0.18682434)	0.52128211 (0.179531660)	0.744	0.51944713 (0.182979141)	0.295	0.50388571 (0.199028218)	0.463
Cyanobacteria	0.00166496 (0.004473048)	0.00400170 (0.012971676)	0.445	0.00057293 (0.001137753)	0.575	0.00127436 (0.002611528)	0.374
Firmicutes	0.43729664 (0.206868391)	0.42945373 (0.190399620)	0.879	0.42817809 (0.192106268)	0.494	0.43764172 (0.211515643)	0.795
Proteobacteria	0.05220132 (0.056037417)	0.02949878 (0.031325947)	0.014	0.03956061 (0.040967824)	0.059	0.04658518 (0.065482081)	0.332
Tenericutes	0.00102377 (0.001891263)	0.00128817 (0.002529497)	0.515	0.00145432 (0.003484704)	0.721	0.00090008 (0.001704735)	0.241
Verrucomicrobia	0.01246812 (0.036769127)	0.01101696 (0.028079550)	0.300	0.00677065 (0.015512495)	0.496	0.00462997 (0.010871109)	0.158
Family (SD)							
Clostridiaceae	0.000972 (0.001774)	0.001339 (0.003118)	0.972	0.001526 (0.002921)	0.778	0.000113 (0.000213)	0.006
Genus (SD)							
*Clostridium* sensu strictu 1	0.00097060 (0.001771992)	0.00132653 (0.003120542)	0.917	0.00151798 (0.002923143)	0.875	0.00011240 (0.000212310)	0.006
*Anaerostipes*	0.00074423 (0.001353722)	0.00356675 (0.005512016)	0.021	0.00109881 (0.002487705)	0.401	0.00127885 (0.002147223)	0.208
Ruminococcaceae NKA214	0.00295456 (0.005078750)	0.00175132 (0.002232046)	1.00	0.00093059 (0.000953027)	0.047	0.00179396 (0.002413138)	0.221
Ruminococcaceae UCG002	0.01773434 (0.023868025)	0.02340332 (0.030784359)	0.064	0.02170348 (0.025077344)	0.027	0.02346318 (0.034270461)	0.196

SD: standard deviation. ^a^ Wilcoxon test.

**Table 3 ijms-24-02720-t003:** Prevalence of side effects during the 6-month follow-up from treatment start with DMF.

Side Effects	Month 1	Month 3	Month 6
Gastrointestinal s.e., *n* (%)	8 (42.1)	7 (36.8)	3 (15.8)
Flushing, *n* (%)	8 (42.1)	10 (52.6)	8 (42.1)
Lymphopenia, *n* (%) *	0 (0)	0 (0)	3 (15.8)

* Lymphocytes < 1000/μL.

**Table 4 ijms-24-02720-t004:** Microbiota composition (expressed as relative abundance and SD) by gastrointestinal side effects during follow-up. The microorganisms in light gray are lower in subjects with side effects, and those in purpleare higher.

Relative Abundance, Mean (SD)	GI Side Effects	No GI Side Effects	*p* ^a^
Month 1			
F. Bacteroidaceae	0.142583 (0.096817)	0.366616 (0.233867)	0.033
g. *Bacteroides*	0.142293 (0.096621)	0.363766 (0.230503)	0.033
Ph. Tenericutes	0.002662 (0.003387)	0.000189 (0.000379)	0.037
F. Streptococcaceae	0.008169 (0.015765)	0.001113 (0.001412)	0.021
g. *Streptococcus*	0.008124 (0.015781)	0.001100 (0.001404)	0.013
g. *Subdoligranulum*	0.025199 (0.024159)	0.007276 (0.008687)	0.026
Month 3			
F. Barnesiellaceae	0.003511 (0.004130)	0.015702 (0.011620)	0.022
g. *Barnesiella*	0.002023 (0.003202)	0.013530 (0.010687)	0.013
g. *Odoribacter*	0.001571 (0.001898)	0.004430 (0.002789)	0.033
F. Clostridiaceae	0.002577 (0.003384)	0.000913 (0.002570)	0.031
g. *Clostridium* sensu strictu 1	0.002556 (0.003399)	0.000912 (0.002568)	0.039
g. *Blautia*	0.004612 (0.003223)	0.002111 (0.002281)	0.028
Month 6			
Ph. Proteobacteria	0.008327 (0.008004)	0.054783 (0.069695)	0.023
F. Acidaminococcaceae	0.001691 (0.002929)	0.026815 (0.041287)	0.017
F. Burkholderiaceae	0.004832 (0.005507)	0.020434 (0.013933)	0.032
g. *Clostridium* sensu strictu 1	0.000416 (0.000361)	0.000047 (0.000098)	0.051

SD: standard deviation; GI: gastrointestinal; Ph: phylum; F: family; g.: genus. ^a^ Kruskal–Wallis test.

**Table 5 ijms-24-02720-t005:** Microbiota composition (expressed as relative abundance and SD) by flushing occurrence during the follow-up. The microorganisms in light gray are lower in subjects with flushing, and those in purple are higher.

Relative Abundance, Mean (SD)	Flushing	No Flushing	*p* ^a^
Month 1			
g. *Paraprevotella*	0.025157 (0.046213)	0.002366 (0.005210)	0.025
Month 3			
g. *Paraprevotella*	0.001683 (0.003071)	0.012232 (0.016627)	0.047
g. *Streptococcus*	0.002681 (0.002109)	0.001039 (0.002010)	0.011
F. Pasteurellaceae	0.000906 (0.001164)	0.000095 (0.000126)	0.029
g. *Haemophilus*	0.000905 (0.001163)	0.000089 (0.000121)	0.024
F. Succinivibrionaceae	0.0 (0.0)	0.023321 (0.046290)	0.054
g. Lachnospiraceae NK4A136 group	0.013219 (0.009539)	0.006049 (0.007685)	0.050
g. *Faecalibacterium*	0.169328 (0.109237)	0.084817 (0.090988)	0.050
Month 6			
F. Enterobacteriaceae	0.000217 (0.000230)	0.004621 (0.010500)	0.021
g. *Akkermansia*	0.000214 (0.000606)	0.007914 (0.014369)	0.036
*g. Lachnospira*	0.013822 (0.012280)	0.002903 (0.004087)	0.011
F. Pasteurellaceae	0.000957 (0.001510)	0.000029 (0.000050)	0.025
g. *Haemophilus*	0.000955 (0.001508)	0.000029 (0.000050)	0.025
g. *Clostridium* sensu strictu 1	0.000221 (0.000274)	0.000016 (0.000048)	0.056

SD: standard deviation; Ph: phylum; F: family; g.: genus. ^a^ Kruskal–Wallis test.

**Table 6 ijms-24-02720-t006:** Studies on the impact of DMTs for MS on the gut microbiota.

Reference	Study Design	Population Size	Geographical Location	Results
Jangi et al. 2016 [9]	Case-control study	60 pwMS (28 non-treated, 18 treated with IFNβ and 14 with GA), 43 HC	USA	Treated MS patients had increased *Prevotella* and *Sutterella*, which were either significantly reduced or showed a trend of reduced populations in untreated patients compared with HC. The genus *Sarcina* was instead reduced in treated patients vs untreated.
Cantarel et al. 2015 [10]	Case-control cross-sectional study and longitudinal prospective study (samples collected at baseline and 90 days after starting vitamin D supplementation)	7 pwRRMS with vitamin D deficiency (5 treated with GA, 2 untreated), 8 HC with vitamin D deficiency. Samples of 4 RRMS patients (2 treated with GA) were available for longitudinal evaluation	USA	Difference between treated with GA and untreated MS patients in the abundance of the family Bacteroidaceae and the genera *Faecalibacterium*, *Ruminococcus*, Lactobacillaceae, and *Clostridium*. GA could affect vitamin D changes in the microbiota: treated MS subjects had increases in *Janthinobacterium* and decreases in *Eubacterium* and *Ruminococcus* after high-dose vitamin D supplementation. Compared to HC and GA-treated MS subjects, untreated MS patients had an increase in the *Akkermansia*, *Faecalibacterium*, and *Coprococcus* genera after vitamin D supplementation.
Tremlett et al. 2016 [11]	Case-control study	18 children ≤18 years old within two years of MS onset (5 treated with GA, 3 with IFNβ, 1 with NTZ), 17 HC	USA, Canada	Treated MS patients showed a greater α biodiversity even if without statistical significance.
Katz Sand et al. 2019 [14]	Cross-sectional study	168 RRMS patients (75 treatment-naïve, 33 treated with DMF, and 60 with GA)	USA	Both therapies were associated with a decreased relative abundance of the Lachnospiraceae and Veillonellaceae families. DMF was associated with a decreased relative abundance of the phyla Firmicutes and Fusobacteria and the order Clostridiales and an increase in the phylum Bacteroidetes.
Storm-Larsen et al. 2019 [15]	Longitudinal prospective pilot study	36 RRMS (27 treated with DMF, 9 with injectable DMTs)	Norway	Trend towards normalization of the low abundance of butyrate-producing *Faecalibacterium* after 12 weeks of treatment. In the DMF patients, there was also a trend of reduced Actinobacteria at two weeks, mainly driven by *Bifidobacterium*.
Diebold et al. 2022 [16]	Longitudinal prospective study	20 RRMS patients treated with DMF	Switzerland	DMF induced no significant changes in the α diversity. Under DMF treatment *Coprococcus eutactus* and *Enterococcus gilvus* were decreased, *Akkermansia muciniphila* was not significantly decreased and *Lactobacillus pentosus* was increased. *A. muciniphila*, *Bacteroides dorei*, *Agathobacter rectale*, *Prevotella copri*, and *P. falseni* discriminated between patients with and without subsequent lymphopenia.
Present study	Longitudinal prospective study	19 RRMS treated with DMF	Italy	Biodiversity showed no changes over 6 months of treatment with DMF. At the genus level, DMF was associated with *Clostridium* decrease after 6 months. In subjects reporting side effects, a higher abundance of *Streptococcus*, *Haemophilus*, *Clostridium*, *Lachnospira*, *Blautia*, *Subdoligranulum*, and Tenericutes and lower of *Bacteroidetes*, *Barnesiella*, *Odoribacter*, *Akkermansia*, and some Proteobacteria families were detected.

DMTs: disease-modifying treatments; MS: multiple sclerosis; pwMS: people with MS; RRMS: relapsing-remitting MS; IFN: interferon; HC: healthy controls; GA: Glatiramer Acetate; NTZ: natalizumab; DMF: dimethyl fumarate.

## Data Availability

The datasets used for this study are available from the corresponding author upon reasonable request.

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
