# Peer review of "Gut Microbiota Changes during Dimethyl Fumarate Treatment in Patients with Multiple Sclerosis"

_ijms, 2023, doi:10.3390/ijms24032720_

Round 1
Reviewer 1 Report (Previous Reviewer 1)
In this longitudinal study, the authors aimed to study the changes in gut microbiota composition during the first 6 months of 19 MS patients
treated with dimethyl fumarate (DMF). The novelty of this study is significant, the article is overall clear, and in my opinion only needs some minor adjustments before publication, as stated below:
1. The authors reported that 7 patients were previously treated with DMTs. It would be of interest in my opinion which DMTs were used in these patients, and a brief discussion about if this could explain the high interivindual variability at baseline (or not).
2. Another useful information that could be added is the previous disease activity and, realted to this consideration, the eventual reason for switching to DMF
3. Did all the patients responded well to DMF treatment?
4. I suggest to expand the limitation section, by discussing also the limitations related ti the study sample (e.g. the small number of recruited participants prevented the authors to use multivariable analyses to detect other factors associated with the explored differences), and the absence of a comparison group.
Author Response
Please see the attachment.

Reviewer 2 Report (New Reviewer)
Dimethyl fumarate (DMF) was approved by the US FDA on 2013 and its generic forms in 2020. The European Medicine Agency approved this compound on 2014 and the generic follow-ons in the summer of 2022. This opens the possibilities of larger populations with MS to be exposed to this medications, ergo, the implications extracted from your study. There are at least 6 other oral medications MS disease modifying therapies, although it is granted that DMF has been previously studied in the context of microbiota composition, (1) it begs the question of specifically why DMF was selected to be studied and not other medication, i.e. S1P1 receptor modulators and teriflunomide.
(2) There are other approved related medicines with similar MOA: Diroximel fumarate and Monomethyl fumarate. These however are not mentioned in the discussion. All, although each of these drugs has a different degree of severity and frequency of side effects, these are common in nature. Would theoretically these related medications have an expected similar side effect in gut microbiota biodiversity?
(3) The notable variability of amounts of paraprevotella in association with flushing during the diverse stages of the study, and the inconsistent presence and association of other microbial families to side effects and lymphocytopenia are not strong enough (in the perception of this reviewer).
(4) It appears you included patients with MS being screened to initiate DMF therapy. This inclusion resulted in the presence in your series of 21% of individuals with other autoimmune disorders, enough proportion to raise a signal on potential influence in the identification of baseline microbiota in your cases. Your comments will be appreciated.
Round 2
Reviewer 2 Report (New Reviewer)
Thank you for addressing this reviewer questions.
This manuscript is a resubmission of an earlier submission. The following is a list of the peer review reports and author responses from that submission.
Round 1
Reviewer 1 Report
In this explorative study, the Authors have analyzed gut microbiota changes during treatment with DMF and TFN in 24 RRMS patients over the short term (up to 6 months after treatment initiation). Moreover, they considered the possible effect of lifestyle factors that could have influenced the observed results. The small sample size is the main limitation of this study, next to the short study duration. Finally, due to the small sample size, no multivariable analyses were possible. Therefore, I would suggest to temper the conclusions of the study in light of these observations. Apart from this minor change and a minor spell check, I have no further comments.
Author Response
Thanks for your comment. As suggested, we have temperated the conclusions. The changes to the text are highlighted in yellow.
Reviewer 2 Report
REVIEW: Investigating the effect of oral disease modifying treatments on the gut microbiota composition in patients with multiple sclerosis.
The present work is a longitudinal study where the effect of dimethyl fumarate and teriflunomide on the gut microbiota as well as lifestyle elements has been evaluated in patients with multiple sclerosis. Although the study, according to the authors, shows evidence not found to date, the work has serious problems. the study has been written in a very confusing way which makes it very difficult to follow the workflow. It requires a complete reworking of the structure, language, and wording. It also contains methodological doubts.
Given that there are only 4 cases with teriflunomide, no conclusions can be derived from such a small sample. the study should focus only on dimethyl fumarate. the other is anecdotal.
ABSTRACT
“Gut microbiota biodiversity and phyla showed no changes over the first 6 months of treatment with DMF nor TFN, pointing to DMT overall safety on microbiota function. As for the potential dysbiotic role of Clostridium subspecies, DMF could contribute to restoring gut microbiota climax at the genera level among smokers and subjects with low sun exposure”. This paragraph is not understandable, it talks about safety in terms of changes in diversity and at the phylum level and then talks about the potential risk of dysbiosis by Clostridium, when it is an essential genus in the intestinal microbiota with several beneficial species and how DMF restores "gut microbiota climax” (?) at the genus level in smokers and with subjects with low sun exposure. it is all very confusing.
TITLE
A more appropriate title should be: The effect of a Disease-Modifying Oral Treatment on Gut Microbiota in patients with Multiple Sclerosis.
INTRODUCTION
Lines 38-42: The gut microbiota has more functions than those mentioned above.
Lines 55-62: Should be placed in a different paragraph. introduce a different item (DMF).
Table 1 is out of place in the introduction. It could be briefly commented that there are several studies that have shown that the gut microbiota in MS patients is different from that of healthy controls, even that it is different in relapses but not with a detailed table. These studies should be commented on in the discussion.
The last paragraph should precisely describe the objective of the study.
This should be followed by the material and methods section to follow the workflow of the study. Presenting the results directly is not well understandable.
MATERIALS AND METHODS
The study design should be more clearly and specifically defined. The article should have followed the STROBE guidelines.
The reference to the McDonald criteria for the diagnosis of MS is missing.
No information on DMF or TFN product: dosage, dosage form, why some patients take one or the other, if it is part of conventional clinical practice.
¿Why are these lifestyle factors studied and not others?
“Bifidobacteria subspecies were analyzed using the internal transcribed spacer bifidobacterial profiling (ITSbp) [32]. The database Silva v. 132 and a customized database containing currently known Bifidobacteria sequences were used to obtain the 16S rRNA analysis outputs and bifidobacterial ITS analysis, respectively”. ¿Why are bifidobacteria studied at the subspecies level? How reliable is this test. ¿Why wasn't a qPCR done?
Line 354: ¿Which paper are you referring to, and did it have the same design as this paper?
Line 348: ¿Which study groups do you consider?
Given that it is a longitudinal, observational study with different measurement points, why was regression of some kind not chosen?
RESULTS
In general, it is very difficult to follow the results. The results should be structured in sub-sections
Lines 76-78: This info should be placed in materials and methods section.
Table 2: Display the continuous variables with their mean and their 95% confidence interval.
In general, the structure of this section needs to be revised, as some points are mixed up with others. More paragraphs should be used as well as sub-sections.
many data that are already shown in the baseline table are commented, so there is no need to comment them again.
Lines 100-103: In Materials and Methods section.
Lines 104-108: In Materials and Methods section commented variables. should be placed in a sub-section on gut microbiota analyses
Line 109: In silico analyses¿?
All these big tables should be included in supplementary material and collect the most relevant data.
¿Only the species B. adolescentis is detected? ¿Were not more to be detected? ¿Were not subspecies mentioned in the material and methods?
¿Are the comparisons made in the different months of the different taxa against baseline or against the previous month? This should be detailed in material and methods.
Line 133: Do not discuss the results in the results section, just state them.
DISCUSSION
The first paragraph repeats what was stated in the intro. the first paragraph of the discussion should assess an overall view of the results.
In general, it is very difficult to follow the discussion section.
Lines 156-158: In the previous sentence you mention the baseline and now that the alpha-diversity and phylum do not change throughout the study. Very confusing.
Line 158: “Our results are in line with previous findins”. ¿Regarding which variables?
Lines 163-165: You should first show your results and discuss them with others in the literature, not the contrary.
Lines 165-167: ¿Who says so? There are studies showing changes in the composition of the gut microbiota throughout the day.
Lines 169-178: Very confusing and poorly worded paragraph.
Lines 182-185: You discusses in vitro studies (without references) but then talks about a murine model.
“Clostridiales may be therefore involved in generating neuroinflammation and –degeneration, and at the same time can act as a potential target for DMTs”. Very confusing
Lines 187-192: It is very difficult to reach that conclusion with what has been developed above. C. perfrigens is not a normal member of the intestinal microbiota and therefore the mechanism is futile in relation to the study.
Lines 196-199: I can't make sense of this sentence.
Line 213: Bifidobacterium.
Lines 214-229: No correlation has been made and therefore the above mentioned conclusions cannot be reached

Round 2
Reviewer 2 Report
Without going into details, the changes introduced do not improve the grammatical, stylistic and methodological deficiencies of the present study. it is very poorly written and this makes it difficult to understand. The main findings are not detailed, at some points there are contradictions. I suggest a complete rewrite to re-evaluate it on my part. The statistical approach should be more consistent. the tables should be better executed. there is no description of the basal microbiota of these patients, etc.
